# Development and characterization of a *Gucy2d-cre* mouse to selectively manipulate a subset of inhibitory spinal dorsal horn interneurons

Elizabeth K. Serafin[1]*, Judy J. Yoo[1,2], Jie Li[1], Xinzhong Dong[3], Mark L. Baccei[1]

**1** Pain Research Center, Department of Anesthesiology, University of Cincinnati Medical Center, Cincinnati, OH, USA, **2** Medical Scientist Training Program, University of Cincinnati, Cincinnati, OH, USA, **3** Departments of Neuroscience, Neurosurgery and Dermatology, Johns Hopkins University School of Medicine, Baltimore, MD, USA

* kritzeee@ucmail.uc.edu

**Data Availability Statement:** All quantification data and image files related to quantification are available from the Open Science Framework

## Abstract

Recent transcriptomic studies identified *Gucy2d* (encoding guanylate cyclase D) as a highly enriched gene within inhibitory dynorphin interneurons in the mouse spinal dorsal horn. To facilitate investigations into the role of the *Gucy2d+* population in somatosensation, *Gucy2d-cre* transgenic mice were created to permit chemogenetic or optogenetic manipulation of this subset of spinal neurons. *Gucy2d-cre* mice created via CRISPR/Cas9 genomic knock-in were bred to mice expressing a cre-dependent reporter (either tdTomato or Sun1. GFP fusion protein), and the resulting offspring were characterized. Surprisingly, a much wider population of spinal neurons was labeled by cre-dependent reporter expression than previous mRNA-based studies would suggest. Although the cre-dependent reporter expression faithfully labeled ~75% of cells expressing *Gucy2d* mRNA in the adult dorsal horn, it also labeled a substantial number of additional inhibitory neurons in which no *Gucy2d* or *Pdyn* mRNA was detected. Moreover, cre-dependent reporter was also expressed in various regions of the brain, including the spinal trigeminal nucleus, cerebellum, thalamus, somatosensory cortex, and anterior cingulate cortex. Injection of AAV-CAG-FLEX-tdTomato viral vector into adult *Gucy2d-cre* mice produced a similar pattern of cre-dependent reporter expression in the spinal cord and brain, which excludes the possibility that the unexpected reporter-labeling of cells in the deep dorsal horn and brain was due to transient *Gucy2d* expression during early stages of development. Collectively, these results suggest that *Gucy2d* is expressed in a wider population of cells than previously thought, albeit at levels low enough to avoid detection with commonly used mRNA-based assays. Therefore, it is unlikely that these *Gucy2d-cre* mice will permit selective manipulation of inhibitory signaling mediated by spinal dynorphin interneurons, but this novel cre driver line may nevertheless be useful to target a broader population of inhibitory spinal dorsal horn neurons.

repository (DOI 10.17605/OSF.IO/G6BV9). All other relevant data are contained within the manuscript.

**Funding:** All work was supported by NS100469, NIH National Institute of Neurological Disorders and Stroke (https://www.ninds.nih.gov/), awarded to MLB. The funders had no role in study design, data collection and analysis, decision to publish, or preparation of the manuscript

**Competing interests:** The authors have declared that no competing interests exist.

## Introduction

It is widely recognized that discrete subpopulations of inhibitory neurons within the spinal dorsal horn regulate distinct, yet overlapping, somatosensory modalities. Dynorphin neurons, marked by expression of prodynorphin (*Pdyn*), are key for the suppression of mechanical pain and itch, as their loss or ablation evokes mechanical allodynia and enhanced scratching in response to a range of pruritogens [1–3]. Importantly, GABAergic synapses originating from spinal dynorphin interneurons provide the majority of inhibitory synaptic input onto lamina I spinoparabrachial projection neurons [4], suggesting that inhibitory dynorphin neurons are key modulators of ascending nociceptive transmission at the level of the spinal cord.

Although dynorphin neurons are easily identified and targeted through the use of an existing *Pdyn-IRES-cre* driver mouse line [5], the dynorphin population in the superficial dorsal horn (SDH) is not solely composed of inhibitory neurons. Excitatory neurons comprise approximately 30% of dynorphin neurons in laminae I-II of the SDH, with additional glutamatergic dynorphin neurons found in lamina III [6, 7], which prevents the straightforward use of this driver line to selectively dissect the role of the inhibitory subset of this mixed population. Indeed, it is likely that the itch-suppressing function of spinal dynorphin interneurons is achieved primarily through GABAergic neurons, while mechanical hypersensitivity may be modulated at least in part by excitatory dynorphin neurons [7]. Moreover, dynorphin-expressing neurons are also found in the dorsal root ganglia (DRG; [8, 9]) and several brain regions [10–12], which necessitates the use of dual-recombinase-dependent intersectional genetic approaches [3, 4] or spatially restricted administration of genetic payloads (e.g., through intraspinal injection of viral vectors; [13, 14]) to selectively manipulate gene expression in inhibitory dynorphin neurons solely at the level of the spinal cord.

*Gucy2d*, encoding the membrane-bound guanylate cyclase D (GC-D), has recently emerged as a highly enriched and highly specific marker of inhibitory dynorphin neurons in the spinal dorsal horn, with no *Gucy2d* mRNA detected in either the dorsal root ganglia (DRG) or brain [15–17]. Previously, *Gucy2d* expression had only been documented in a small population of olfactory sensory neurons in the cul-de-sacs of caudal olfactory turbinates [18–20] and in axons originating from these neurons which project to the necklace glomeruli in the caudal olfactory bulb [19, 21, 22]. In the main olfactory epithelium (MOE), GC-D may be involved in carbon dioxide detection [20, 23] or socially transmitted food preference [24, 25], but its function in the spinal cord is not known. Nevertheless, 94% of *Gucy2d*-expressing cells in the SDH co-express *Pdyn* mRNA, and virtually all express the inhibitory marker gene *Pax2* [17]. Moreover, *Gucy2d*-expressing cells make up about half of dynorphin-lineage neurons in lamina I-III [15] and thus account for a substantial proportion of inhibitory dynorphin neurons [6]. This raises the possibility that *Gucy2d* may provide an avenue to genetically target the inhibitory subpopulation of spinal dynorphin neurons in a straightforward manner, thereby permitting chemogenetic or optogenetic manipulation of these cells with the goal of elucidating their role in modulating pain and/or itch while eliminating unwanted off-target effects of gene expression in other areas of the central and peripheral nervous systems.

Here, we describe a novel *Gucy2d-cre* driver mouse line that was designed to permit the selective targeting and manipulation of spinal inhibitory dynorphin neurons. *Gucy2d-cre* mice created via CRISPR/Cas9 genomic knock-in were bred to mice expressing a cre-dependent reporter (either tdTomato or nuclear Sun1.GFP fusion protein), and the resulting offspring were characterized. We found that this driver line labels a subpopulation of inhibitory neurons largely, but not exclusively, located in laminae I-III of the SDH while sparing the DRG, although the majority of these spinal neurons did not co-express *Gucy2d* mRNA. Moreover, cre-dependent reporter expression was observed in several brain regions, which dramatically

reduces the utility of this line in terms of spatial selectivity for the SDH. Although these results preclude the use of this novel line as tool to selectively manipulate inhibitory dynorphin neurons at the level of the spinal cord, *Gucy2d-cre* mice may nevertheless prove useful in other applications within the neuroscience field.

## Methods

### Animals

All animal experiments were performed in accordance with Institutional Animal Care and Use Committee policies at the University of Cincinnati and Johns Hopkins University. The protocol was approved by the Institutional Animal Care and Use Committee of the University of Cincinnati (Protocol Number 23-02-24-02). All efforts were made to minimize suffering.

*Gucy2d-cre* mice were generated at the Johns Hopkins Transgenic Mouse Core using CRISPR/Cas9 genomic editing. Single-cell C57BL/6J embryos were injected with CRISPR reagents including Cas9 mRNA, sgRNA with sequence 5'-GCAGACTCACCTGCCATGATG GG-3', and *Gucy2d-cre* repair plasmid. The chosen sgRNA induced a double-stranded break at the start codon of the *Gucy2d-cre* open reading frame (ORF) at the end of Exon 1 of the *Gucy2d* gene. The *Gucy2d-cre* repair plasmid encoded cre recombinase and B-globin polyA signal flanked by 200-base pair homology arms to encourage homology-directed repair at the site of the double-stranded break, thereby knocking cre recombinase into the *Gucy2d* ORF. The injected embryos were implanted into pseudopregnant ICR females, and samples from the resulting offspring (i.e., potential founders) were genotyped using primers F-CTCTCTCGTGTGGATCCCCA and R-TGCATCGACCGGTAATGCAG to confirm correct insertion of cre recombinase. Cre-positive founders with correct insertion were bred with mice homozygous for either cre-dependent tdTomato expression (Ai9 mice; Jackson stock #007909) or cre-dependent Sun1.GFP expression (Jackson stock #021039) to facilitate validation and characterization of *Gucy2d-cre* expression using either of these fluorescent reporters. Cre-negative Ai9 and Sun1.GFP mice (n = 3 Ai9; n = 3 Sun1.GFP) did not exhibit reporter expression in the lumbar spinal cord or brain, indicating a lack of cre-independent (i.e., "leaky") expression (S1 Fig A-H). *Gucy2d-cre$^{+/-}$;Rosa26-LSL-tdTomato$^{+/-}$* offspring are referred to as *Gucy2d-tdTomato* mice, and *Gucy2d-cre$^{+/-}$;Rosa26-LSL-Sun1.GFP$^{+/-}$* offspring are referred to as *Gucy2d-Sun1.GFP* mice. Although four cre-positive potential founder *Gucy2d-cre* mice (2 male and 2 female) and their progeny were analyzed, one single male founder was ultimately selected for further characterization in this study. All data and images in this manuscript are derived from that single founder and its progeny.

### Tissue preparation, immunohistochemistry, and in situ hybridization

Adult (8–12 weeks) *Gucy2d-tdTomato* or *Gucy2-Sun1.GFP* mice of either sex were euthanized via sodium pentobarbital overdose and transcardially perfused with 0.1M phosphate buffer (PB) followed by 4% paraformaldehyde (PFA) in PB. Dorsal root ganglia (DRG) were postfixed for an additional hour in 4% PFA, lumbar spinal cords were postfixed for an additional 2 hours, and brains were postfixed for an additional 6 hours. For main olfactory epithelium, noses were stripped of extraneous skin and muscle tissue, then decalcified in 0.5M EDTA for 3 nights at 4˚C. Fixed tissue was transferred to 30% sucrose in RNAse-free 0.01M phosphate-buffered saline (PBS) and stored overnight at 4˚C. Tissue sections were cut on a Leica 1860CM cryostat and mounted on SuperFrost Plus slides (Fisher). *Gucy2d-tdTomato* mice were used for anatomical characterization of sites of *Gucy2d-cre* expression, while *Gucy2d-Sun1.GFP* mice were used for *in situ* hybridization and immunohistochemistry experiments owing to easier visualization of the nuclei of cre-expressing cells.

*In situ* hybridization experiments on spinal cord tissue sections obtained from *Gucy2d-Sun1.GFP* mice were carried out using RNAScope Multiplex Fluorescent Kit v2 (Advanced Cell Diagnostics) according to manufacturer's directions. RNAScope probes for *Gucy2d* (425451-C2), *Slc17a6* (319171-C3), and *Pdyn* (318771) were used with TSA Plus Cyanine 3 and Cyanine 5 systems (Perkin Elmer) for visualization. Following the RNAScope protocol, immunostaining for GFP (ThermoFisher anti-GFP #A10262; 1:500) was carried out to enhance visibility of Sun1.GFP-tagged nuclei. In a separate set of experiments, spinal cord tissue sections obtained from *Gucy2d-Sun1.GFP* mice were stained for NeuN (NovusBio anti-Rbfoxp3 #NBP1-92693; 1:500).

## Image acquisition and analysis

Images were captured on a BZ-X810 inverted fluorescent microscope (Keyence) or a BX63 upright fluorescent microscope (Olympus) using CellSens Dimension Software (Olympus). Images obtained under 20X or 40X magnification were acquired as Z-stack images and projected as Extended Focal Images/Full-Focus Images. Lower magnification images were acquired using 4X or 10X magnification and a single focal plane. For quantitative *in situ* and immunohistochemistry experiments, 3–4 non-adjacent lumbar spinal cord sections from each of 4 mice were evaluated. Cells were considered to be positive for a given *in situ* hybridization target only if 3 or more puncta in the appropriate fluorescent channel were detected within or touching the boundary of the GFP-labeled nucleus of that cell. DAPI was used as an additional nuclear stain. Full quantification data and images are deposited in the Open Science Framework repository at DOI 10.17605/OSF.IO/G6BV9.

## Administration of cre-dependent AAV viral vectors

Adult *Gucy2d-cre* mice of either sex were either injected intraspinally with AAV8-CAG-FLEX-tdTomato (Addgene # 8306-AAV8; n = 4 cre-positive mice; n = 2 cre-negative controls) or intravenously with AAV-PHP.eB-CAG-FLEX-tdTomato (Addgene # 8306-PHPEB; n = 3 cre-positive mice; n = 2 cre-negative controls) under isoflurane anesthesia (2.5%). For intraspinal injections, two 300 nl injections of AAV8-CAG-FLEX-tdTomato (titre adjusted to $10^{13}$ gc/ml with sterile PBS) were delivered to the left spinal dorsal horn at levels L3 and L4 at a rate of 60 nl/min, as previously described [26, 27]. For intravenous administration, $10^{11}$ gc of AAV-PHP.eB-CAG-FLEX-tdTomato were diluted in 100 μl of sterile 0.9% saline and injected into the retro-bulbar sinus [28]. For all experiments involving viral vectors, tissue was harvested 3 weeks after injection and processed as described in section 2.2.

## Results

*Gucy2d-cre* mice were generated by using CRISPR/Cas9 genomic editing to insert DNA encoding cre recombinase at the start codon of the *Gucy2d* open reading frame located at the end of Exon 1 (Fig 1A). C57BL/6J zygotes were injected with CRISPR components including mRNA encoding Cas9, guide RNA (sgRNA), and *Gucy2d-cre* repair plasmid containing sequences for cre recombinase and B-globin polyA signal flanked by 200-bp homology arms to facilitate homology-directed repair of the induced double-stranded break. The zygotes were then implanted into pseudopregnant dams. Since the pronuclear injection of CRISPR components was performed at the single-cell stage, the resulting offspring were not chimeric and were therefore genotyped for the presence of the *Gucy2d-cre* insertion sequence. On this basis, two male and two female potential founders carrying the *Gucy2d-cre* allele were bred to Ai9 reporter mice, which express tdTomato in a cre-dependent manner. Three out of the four potential founder mice produced *Gucy2d-tdTomato* offspring with no obvious differences in

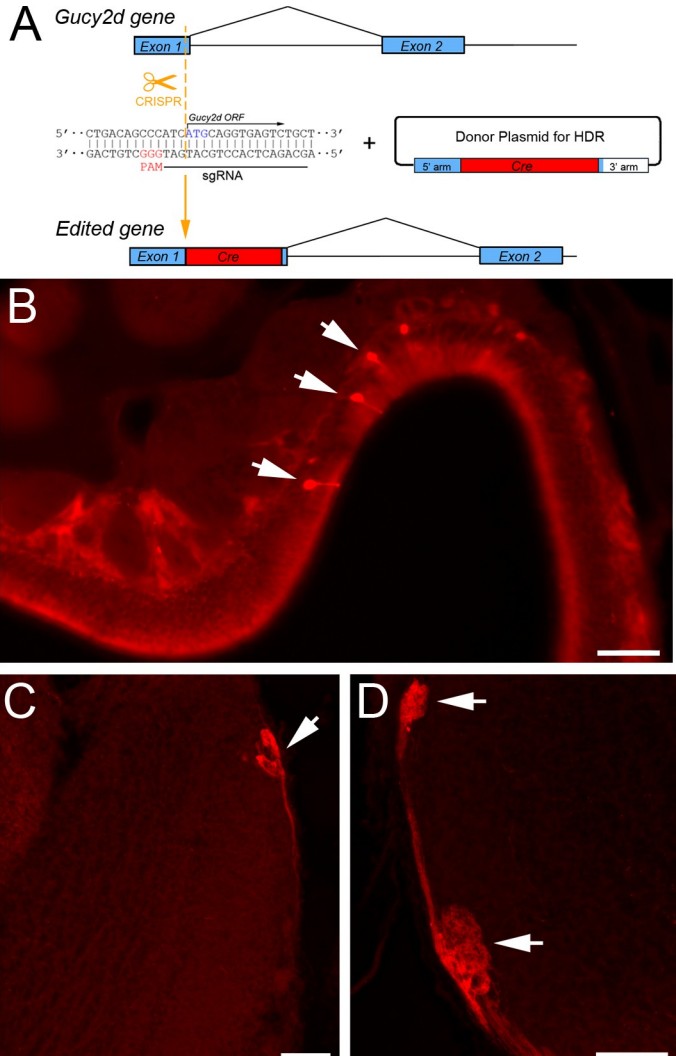

**Fig 1. Generation of *Gucy2d-cre* mice via CRISPR knock-in.** (A) sgRNA was designed to target the start codon (blue text) of the *Gucy2d* open reading frame (ORF), which starts at nucleotide 95 of Exon 1. A DNA sequence encoding Cre recombinase flanked by 200-bp homology arms was inserted at the double-stranded break (dashed line) through homology-directed repair. Thus, Cre recombinase is knocked into Exon 1 of the *Gucy2d* gene. PAM, protospacer-adjacent motif. (B) *Gucy2d-tdTomato* mice exhibit cre-dependent reporter expression in olfactory neurons located in the cul-de-sac regions of the main olfactory epithelium (MOE), as would be predicted based on prior studies of *Gucy2d* expression. Scale bar = 50 μm. (C, D) Similarly, tdTomato+ axons originating from MOE neurons terminate in the necklace glomeruli in the caudal olfactory bulb. Scale bar in panel C = 100 μm; D = 50 μm.

cre-dependent reporter expression. The fourth founder, a male, produced offspring which were highly variable in terms of cre-dependent reporter expression, with some exhibiting expression that was largely similar to the offspring of the other founders while their littermates exhibited little to no tdTomato expression despite carrying the *Gucy2d-cre* mutant allele. Due to this inconsistency, this founder and its offspring were excluded from the study. In all other *Gucy2d-tdTomato* mice examined, the pattern of expression was highly consistent and showed no obvious differences between male and female littermates nor any differences based on whether the sex of the cre-donating parent was male or female. A single founder male was

selected to establish the line, and all data shown and described in this manuscript are derived from the progeny of that founder.

The main olfactory epithelium (MOE), spinal cord, brain, and dorsal root ganglia (DRG) of the resulting *Gucy2d-tdTomato* mice were assessed by histology to determine whether cre-dependent tdTomato expression was consistent with previous characterizations of *Gucy2d* mRNA or its encoded protein guanylate cyclase D (GC-D) expression in these tissues [15, 17–19]. As expected, *Gucy2d-tdTomato* mice exhibit cre-dependent reporter expression in a small number of neurons in MOE turbinates (Fig 1B), and tdTomato-labeled glomeruli were also observed in the caudal olfactory bulb, consistent with the location of necklace glomeruli (Fig 1C–1D).

A population of cells in the spinal superficial dorsal horn (SDH) of *Gucy2d-tdTomato* mice was labeled with cre-dependent reporter expression at the cervical, thoracic, and lumbar levels (Fig 2A–2C), which was consistent with previous studies in which *Gucy2d* mRNA was detected in a subset of spinal dynorphin-lineage SDH neurons [15, 17]. Also consistent with previous studies, no cre-dependent reporter expression was observed in the DRG (Fig 2D).

To characterize the SDH cells labeled by *Gucy2d-cre* dependent reporter expression, *Gucy2d-cre* mice were bred to mice expressing a Sun1-GFP fusion protein in a cre-dependent

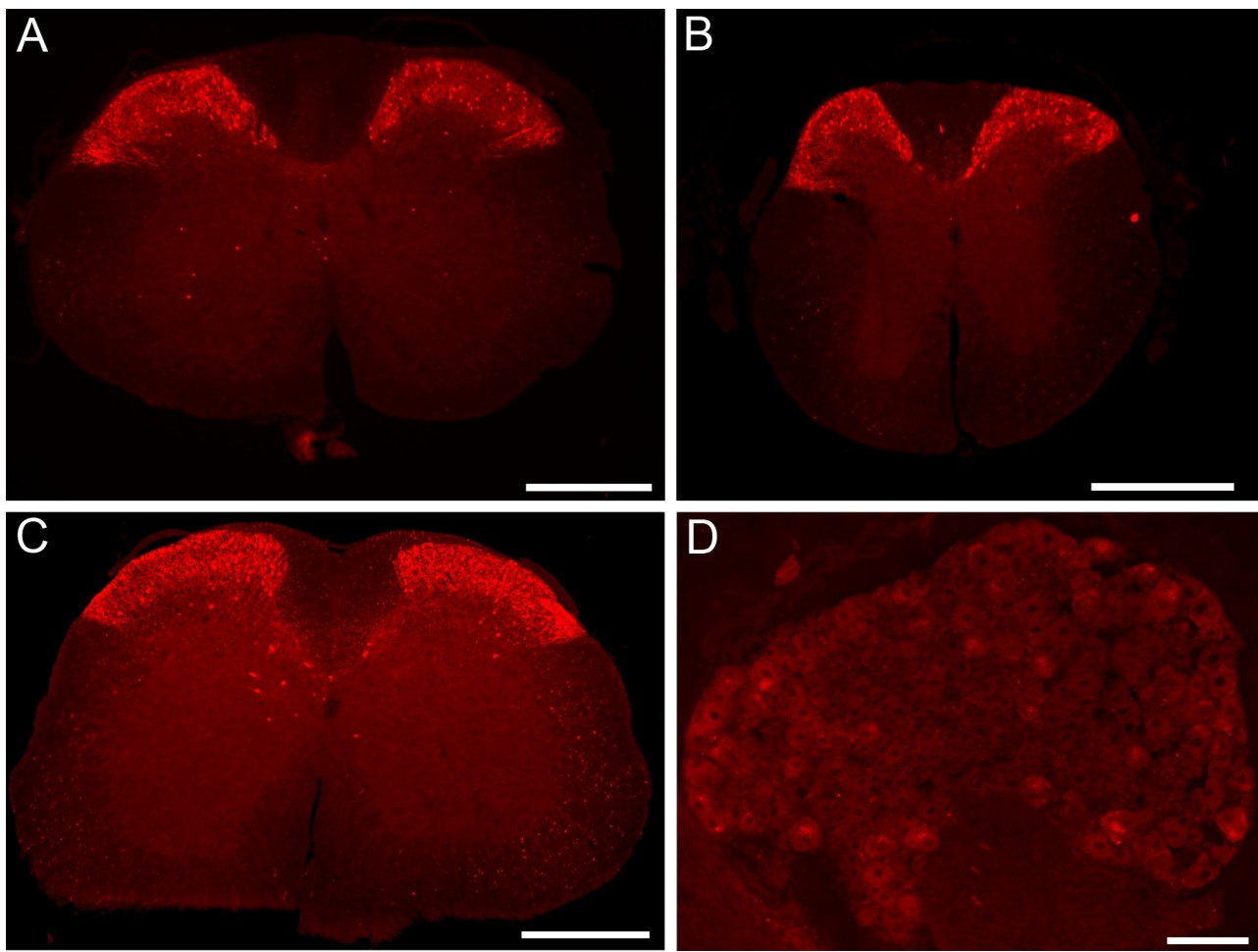

**Fig 2. *Gucy2d-cre* driven reporter expression labels a subset of cells in the spinal dorsal horn while sparing the dorsal root ganglia.** *Gucy2d-tdTomato* mice exhibit cre-dependent reporter expression in the superficial dorsal horn of the cervical (A), thoracic (B) and lumbar (C) spinal cord, which is most prevalent in laminae I-III. Scale bar in panels A-C = 500 μm. (D) Cre-dependent reporter expression is absent from the dorsal root ganglia (DRG) of *Gucy2d-tdTomato* mice. Scale bar = 100 μm.

manner, such that the nuclei of cre-expressing cells (or cells that had previously expressed cre at one point in development) were labeled with GFP. Our earlier examinations of cre-dependent reporter expression in *Gucy2d-tdTomato* mice, in which the morphology of cells could be more clearly visualized, indicated that the population of spinal cord cells labeled by the *Gucy2d-cre* mouse line was likely neuronal; however, we confirmed this by performing immunohistochemistry for the neuronal marker NeuN. Indeed, NeuN staining was present in 93.03% ± 1.05% of *Gucy2d-Sun1.GFP* nuclei (Fig 3A and 3D). Next, to determine whether the labeled neurons were excitatory or inhibitory, we performed *in situ* hybridization for *Slc17a6*, which encodes the vesicular glutamate transporter VGLUT2 and is expressed by virtually all excitatory neurons in the SDH [29, 30]. *Gucy2d-Sun1.GFP* cells were overwhelmingly inhibitory, with only 7.49% ± 0.99% co-expressing *Slc17a6* (Fig 3B and 3D). Finally, we performed *in situ* hybridization experiments to quantify the colocalization of *Gucy2d* and/or *Pdyn*

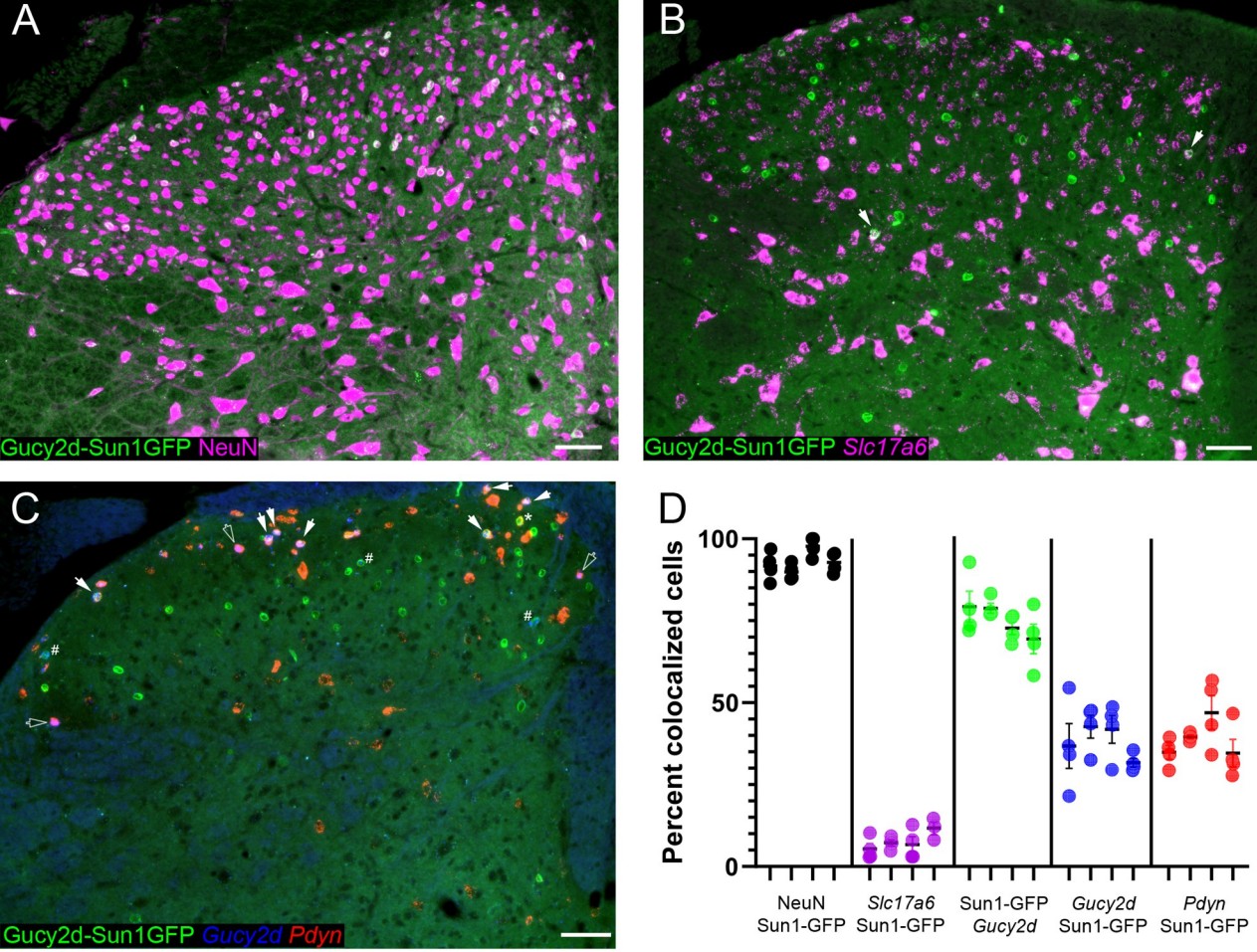

**Fig 3. Characterization of SDH neurons labeled by *Gucy2d-cre* reporter expression.** (A) Reporter-labeled cells in the SDH of *Gucy2d-Sun1.GFP* mice (green) exhibit high colocalization with neuronal marker NeuN (magenta; 93.03% ± 1.05%). (B) Only a small number of *Gucy2d-Sun1.GFP* cells express *Slc17a6* mRNA (filled arrow), encoding the glutamatergic marker VGLUT2 (magenta; 7.49% ± 0.99%). (C) Although most cells expressing *Gucy2d* mRNA are labeled by *Gucy2d-cre* reporter expression (75.08% ± 1.90%), only a minority of the total population of reporter-labeled cells express *Gucy2d* mRNA (blue; 38.27% ± 2.28%). A similar percentage of reporter-labeled cells express *Pdyn* mRNA (red; 39.00% ± 2.03%). Filled arrows: *Gucy2d-Sun1.GFP* cells expressing both *Gucy2d* and *Pdyn* mRNA. Open arrows: cells expressing both *Gucy2d* and *Pdyn* mRNA but not labeled with *Gucy2d-Sun1.GFP*. Asterisk (*): *Gucy2d-Sun1.GFP* cells expressing *Pdyn* but not *Gucy2d* mRNA. Pound sign (#): *Gucy2d-Sun1.GFP* cells expressing *Gucy2d* but not *Pdyn* mRNA. Scale bars in A-C = 50 µm. (D) Quantification of *in situ* hybridization data shown in (A-C). N = 4 for all experiments; each tick on x-axis indicates one biological replicate, and each data point indicates one tissue section.

(encoding preprodynorphin) mRNA within the *Gucy2d-Sun1.GFP* population in the dorsal horn. *Gucy2d* mRNA was strongly expressed in a subpopulation of cells located in lamina I and II of the SDH, with a few scattered cells located in lamina III. 75.08% ± 1.90% of cells expressing *Gucy2d* mRNA exhibited Sun1.GFP+ nuclei, indicating that most of the *Gucy2d*-expressing population is captured by this mouse line (Fig 3C and 3D). However, while *Gucy2d* mRNA was mainly localized to lamina I and the superficial portion of lamina II, a substantial number of *Gucy2d-Sun1.GFP* nuclei were clearly present in lamina III and deeper dorsal horn laminae, where no *Gucy2d* mRNA was detected. Accordingly, *Gucy2d* mRNA was detected in only a minority of the SDH cells labeled by this mouse line (38.27% ± 2.28%; Fig 3C, blue; Fig 3D). A similar percentage of *Gucy2d-Sun1.GFP* cells (39.00% ± 2.03%; Fig 3D) express *Pdyn* mRNA (Fig 3C, red), indicating that the GFP-labeled cells lacking *Gucy2d* mRNA are not simply part of the wider dynorphin lineage. Approximately 60% of *Gucy2d-Sun1.GFP* cells in the SDH are clearly lacking mRNA of either target, and the molecular identity of these cells remains unknown (Fig 3C, green).

As is common with breeding strategies involving a cre driver mouse line paired with a cre-dependent reporter mouse, we hypothesized that this seemingly extraneous population of cells labeled by the *Gucy2d-cre* mouse line was due to transient expression of *Gucy2d* (and therefore cre recombinase) during development, leading to reporter expression which persisted even after *Gucy2d* was no longer actively transcribed. Thus, the cells labeled by cre-dependent reporter in *Gucy2d-tdTomato* (Fig 4A) and *Gucy2d-Sun1.GFP* mice compose the full *Gucy2d* lineage, rather than reflecting active *Gucy2d* expression. To test this hypothesis, we injected an AAV8-CAG-FLEX-tdTomato viral vector expressing tdTomato in a cre-dependent manner into the dorsal spinal cords of adult *Gucy2d-cre* mice. Three weeks later, tdTomato expression was present in a pattern similar to that which was achieved when breeding *Gucy2d-cre* mice to a cre-dependent reporter mouse line, with reporter expression clearly present in lamina III and in deeper laminae of the spinal dorsal horn (Fig 4B). This suggests that this deeper population of cells is likely still expressing *Gucy2d-cre* in adulthood (albeit at levels too low to detect with *in situ* hybridization) which is sufficient to induce recombination and thereby permit virally driven cre-dependent reporter expression. Cre-negative mice injected with the same viral vector did not exhibit tdTomato expression in the spinal cord (S2 Fig A).

### *Gucy2d-cre x Ai9 (Rosa26-LSL-tdTom)*        *Gucy2d-cre* + AAV8-FLEX-tdTom

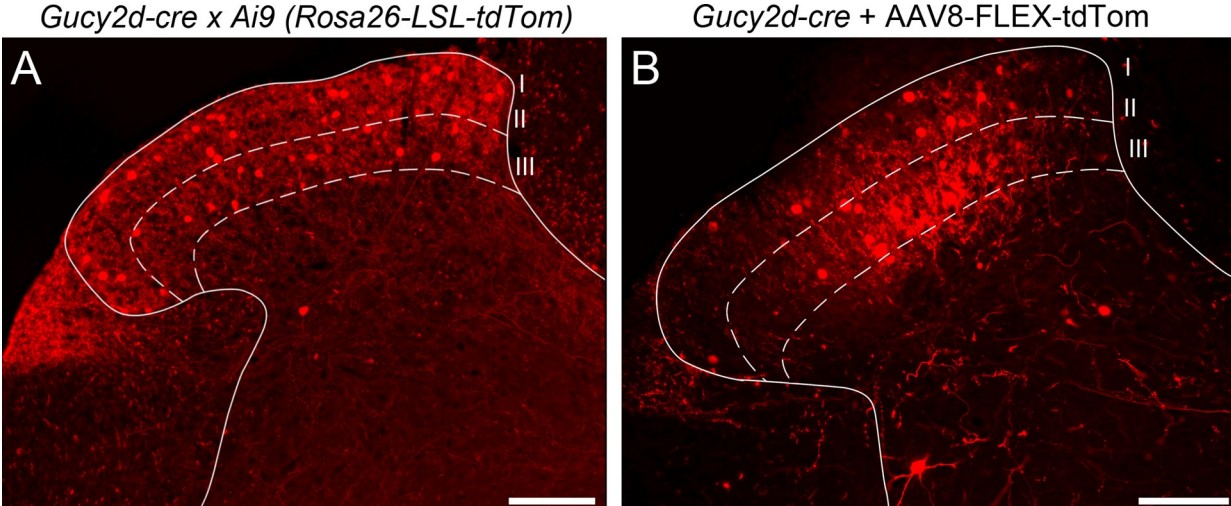

**Fig 4. Cre-dependent reporter labeling in the deeper dorsal horn is not due to transient cre expression during development.** (A, B) The pattern of tdTomato expression driven by *Gucy2d-cre* when crossed to the Ai9 reporter mouse line (A) was similar to tdTomato expression driven by *Gucy2d-cre* after intraspinal injections of AAV8-CAG-FLEX-tdTomato viral vector in adulthood (B). In both cases, tdTomato+ neurons are visible in laminae III and in deeper dorsal horn laminae. Scale bars in panels A-B = 200 μm.

Finally, brains of *Gucy2d-tdTomato* mice were cryosectioned and examined to verify a lack of cre-dependent reporter expression that was expected based on previous mRNA studies [17]. We observed tdTomato-labeled cells in the spinal trigeminal nucleus (Sp5) and, surprisingly, in other brain sites including the cerebellum, thalamus, somatosensory cortex, and anterior cingulate cortex (ACC; Fig 5A–5D). The ACC only exhibited a small number of tdTomato-expressing cells, but expression in the cerebellum, thalamus and somatosensory cortex was quite widespread. To determine whether the cre-dependent reporter expression seen in the brain was the result of transient *Gucy2d* expression earlier in life, adult *Gucy2d-cre* mice were intravenously administered the cre-dependent viral vector AAV-PHP.eB-CAG-FLEX-tdTomato. Three weeks later, cre-dependent tdTomato expression was observed in the same brain regions identified in *Gucy2d-tdTomato* mice (Fig 5E–5H). These results suggest that, similar to

**Fig 5. *Gucy2d-cre* dependent reporter expression does not spare the brain and is not due to transient cre expression during development.** *Gucy2d-tdTomato* mice exhibit widespread expression in multiple brain areas, including the cerebellum and spinal trigeminal nucleus (Cere and Sp5; A), thalamus (Th; B), somatosensory cortex (S1; C), and anterior cingulate cortex (ACC; D). Scale bars in A-B = 1 mm, scale bars in C-D = 500 μm. When adult *Gucy2d-cre* mice were intravenously administered an AAV-PHP.eB-CAG-FLEX-tdTomato viral vector, tdTomato+ cells were observed in the same areas (E, spinal trigeminal nucleus; F, thalamus; G, somatosensory cortex; H, ACC). Scale bars in panels E-G = 500 μm. Other labeled landmarks include 3V: third ventricle, LV: lateral ventricle, M2: secondary motor cortex, PAC: parietal association cortex.

our findings in the spinal cord, the presence of cells labeled by *Gucy2d-cre* despite lacking detectable *Gucy2d* mRNA is not explained by the transient developmental expression of *Gucy2d*, but rather to low levels of active expression in adulthood. Cre-negative mice intravenously administered the same viral vector did not exhibit tdTomato expression in any of these brain regions (S2 Fig B-F).

## Discussion

Inhibitory spinal cord neurons expressing the opioid peptide dynorphin are key inhibitory modulators of lamina I spinoparabrachial projection neurons [4, 31]. As such, they are well positioned to dampen ascending nociceptive transmission, and have indeed been shown to play a role in the suppression of mechanical pain and itch [1–4]. However, dynorphin neurons in the spinal dorsal horn (SDH) are not a homogeneous population and may in fact be divided into multiple subpopulations with distinct molecular phenotypes [16]. At the most fundamental level, spinal dynorphin neurons are not even uniformly inhibitory, with a substantial glutamatergic subpopulation composing up to 30% of the total dynorphin population in the lumbar spinal cord [6]. This poses experimental challenges to targeting inhibitory dynorphin neurons for the purpose of investigating their contributions to nociceptive or pruriceptive processing.

Despite the remarkable selectivity of *Gucy2d* mRNA expression for spinal dynorphin neurons, *Gucy2d* knockout mice do not exhibit altered behavioral responses to nociceptive or pruriceptive stimuli under naïve conditions [17]. Moreover, putative GC-D receptor ligands guanylin, uroguanylin, $CO_2$ and $CS_2$ have been explored in the context of olfaction and socially transmitted food preference [20, 23, 24, 32], but which of these ligands–if any–are operant at the level of the spinal cord is not known. However, we hypothesized that the population of cells which express this gene could nevertheless be involved in modulating the activity of ascending lamina I spinoparabrachial neurons, and thereby pain and itch, through traditional GABAergic inhibitory mechanisms that do not depend on the activities of the GC-D receptor itself.

Multiple genetic strategies have previously been employed to target inhibitory dynorphin neurons in the spinal dorsal horn, which yielded distinct yet overlapping effects on somatosensation. For example, the genetic intersection of the *Lbx1* and *Pdyn* populations selectively targets inhibitory dynorphin neurons throughout lamina I-III of the spinal cord while sparing the DRG and most brain regions, and the ablation of this population induced static and dynamic mechanical allodynia but not itch [3]. However, as ablation was carried out in adulthood, this approach likely also ablates neurons which transiently expressed *Pdyn* earlier in development, thus complicating the assignation of the observed phenotype specifically to adult dynorphin-expressing neurons. Meanwhile, a genetic model which prevents the development of inhibitory spinal dynorphin neurons by suppression of the transcription factor *Bhlhb5* induces spontaneous itch as well as enhanced mechanical sensitivity [1]. It is notable that the spinal neurons which fail to develop in the *Bhlhb5* knockout model, termed "B5-I neurons," are located mainly in laminae I-II. The dorsoventral location of the *Pdyn* neuronal population targeted by a given genetic strategy likely influences the somatosensory phenotypes which are observed following the removal of the *Pdyn* neurons from the SDH network, as prior reports show that *Pdyn* neurons located deeper in the dorsal horn (i.e. along the lamina II-III border) receive a different pattern of primary afferent input compared to *Pdyn* neurons residing in laminae I-II [3]. The selective localization of *Gucy2d* mRNA to inhibitory dynorphin-expressing neurons located overwhelmingly in laminae I-II [15] raised the possibility that a *Gucy2d-cre* driver mouse could be used to more precisely interrogate the role of superficial dynorphin neurons in somatosensation by providing a means to reversibly manipulate this population using optogenetics or chemogenetics.

As a result, a *Gucy2d-cre* mouse was developed to facilitate the straightforward genetic manipulation of spinal inhibitory dynorphin neurons. Cre-dependent reporter expression in the *Gucy2d-tdTomato* offspring of these mice coincided with the previously documented location and distribution of GC-D+ olfactory sensory neurons [18, 19, 21, 22], as tdTomato+ cell bodies were observed in the MOE along with tdTomato+ axonal glomeruli in the caudal olfactory bulb (Fig 1). Similarly, the lack of cre-dependent reporter expression in the DRG of *Gucy2d-tdTomato* mice was also consistent with the lack of *Gucy2d* mRNA expression as shown by *in situ* hybridization experiments [17]. Initial evaluation of spinal cord cre-dependent reporter expression also seemed promising, as *Gucy2d-tdTomato* cells were observed mainly in the SDH (Fig 2), corresponding to the location of most spinal dynorphin neurons and previously documented *Gucy2d* mRNA expression [6, 15].

Unfortunately, further characterization of the cells labeled by cre-dependent reporter expression in the spinal cords of *Gucy2d-Sun1.GFP* mice revealed that, in addition to ~75% of *Gucy2d*-expressing cells in the SDH, this driver line also induces cre-dependent recombination in a substantial population of cells in laminae I-III in which no *Gucy2d* mRNA was detected. *Pdyn* mRNA was also absent from these *Gucy2d*-negative/Sun1.GFP-positive cells, leading to the conclusion that the majority of spinal cells labeled by the *Gucy2d-cre* driver mouse are in fact outside of both the dynorphin and *Gucy2d*-expressing populations (Fig 3). The precise molecular identity of these labeled cells lacking *Gucy2d* and *Pdyn* mRNA is unknown, although the wide majority are inhibitory neurons. To further characterize this population, GFP+ nuclei could be selectively isolated from the lumbar spinal cords of *Gucy2d-cre-Sun1.GFP* mice using the INTACT method [15, 33, 34] and sequenced using single-nucleus RNA-sequencing [16]. On the basis of this study, we would expect this approach to reveal *Pdyn*-positive and *Pdyn*-negative subpopulations within the *Gucy2d-cre* lineage and elucidate the relationship between the *Gucy2d-cre* population and other inhibitory neuronal populations previously identified in the lumbar spinal cord.

The unexpectedly widespread *Gucy2d-cre* expression observed in the SDH contraindicate the use of this driver line in experimental approaches utilizing a simple strategy of breeding the cre driver line to a cre-dependent gene of interest for selective expression in spinal inhibitory dynorphin neurons. Moreover, reporter expression was observed in brain regions including the spinal trigeminal nucleus, cerebellum, thalamus, somatosensory cortex, and anterior cingulate cortex. Experiments utilizing injection of a cre-dependent AAV viral vector produced a similar pattern of reporter expression, indicating that the unanticipated labeling of cell populations in the deep dorsal horn and the brain is not due to transient expression of *Gucy2d-cre* at some early point in development (Figs 4 and 5).

Since this mouse line was developed to facilitate the manipulation of inhibitory spinal dynorphin neurons while sparing the DRG and brain, the supraspinal expression is problematic. The thalamus is prominently involved in somatosensory processing, including the transmission of nociceptive signals to other areas in the brain which ultimately integrate such information into the perception of pain [35, 36]. Moreover, several cortical areas also contained cells labeled by *Gucy2d-cre*, including the primary somatosensory cortex (S1) and anterior cingulate cortex (ACC). These areas not only receive input from the thalamus [37], but also play distinct roles in modulating the sensory-discriminative aspect or affective valence of pain, respectively [38–40]. The *Gucy2d-cre* expression found in these brain regions adds another layer of potential off-target effects on somatosensation. Experiments using this *Gucy2d-cre* driver mouse to investigate the contributions of spinal inhibitory dynorphin neurons to pain or itch behaviors would therefore be confounded by unwanted cre-dependent expression in key areas relevant to somatosensory processing in addition to the spinal cord neurons which are the intended targets of manipulation.

The *Gucy2d-cre* driver line created and described in this study failed to provide a straightforward means to target and manipulate spinal inhibitory dynorphin interneurons, but it may nevertheless be useful for other lines of investigation which seek to target a wider subset of inhibitory spinal cord neurons. Selectivity for the spinal cord may be enhanced through the use of a Flp recombinase driver mouse line in conjunction with dual-recombinase-dependent gene expression. Additionally, the observed pattern of cre-dependent reporter expression in the MOE and olfactory glomeruli suggest that this driver line may have utility for future investigations into how the unique population of olfactory sensory neurons which express GC-D regulate chemosensory processing in vivo.

## Supporting information

**S1 Fig. Ai9 and Sun1-GFP reporter mouse lines do not exhibit cre-independent "leaky" reporter expression.** TdTomato expression is absent from the spinal cords of cre-negative Ai9 mice (A-B), and Sun1-GFP expression is absent from the spinal cords of cre-negative Sun1-GFP mice (C-D). Scale bars in A and C = 200 μm. Scale bars in B and D = 50 μm. TdTomato expression was likewise absent from the brains of cre-negative Ai9 mice, including the areas where *Gucy2d-cre* induced reporter expression in cre-positive mice: spinal trigeminal nucleus, cerebellum (Sp5, Cere; E), thalamus (Th; F), somatosensory cortex (S1; G), and anterior cingulate cortex (ACC; H). Scale bars in E-H = 500 μm. Other labeled landmarks include 3V: third ventricle, LV: lateral ventricle, M2: secondary motor cortex.
(TIF)

**S2 Fig. Viral vector-driven reporter expression is absent in the spinal cord and brain of cre-negative control mice.** (A) Although autofluorescence is visible along the injection tract, intraspinal injection of AAV8-FLEX-CAG-tdTomato to cre-negative mice did not produce viral-driven reporter expression in dorsal horn neurons. Scale bar = 200 μm; dotted line = outline of dorsal horn. (B-F) Intravenous administration of AAV-PHP.eB-CAG-FLEX-tdTomato viral vector to cre-negative mice did not produce tdTomato labeling in the brain. Scale bars in B,D, E, F = 500 μm; Scale bar in C = 200 μm. Brain region and landmark abbreviations include cerebellum (Cere; B), spinal trigeminal nucleus (Sp5; B and C), thalamus (Th; D), somatosensory cortex (S1; E), anterior cingulate cortex (ACC; F), 3V: third ventricle, LV: lateral ventricle, M2: secondary motor cortex.
(TIF)

## Acknowledgments

The authors gratefully acknowledge the Johns Hopkins University Transgenic Mouse Core, including Core Manager Chip Hawkins.

## Author Contributions

**Conceptualization:** Elizabeth K. Serafin, Xinzhong Dong, Mark L. Baccei.

**Data curation:** Elizabeth K. Serafin, Judy J. Yoo.

**Formal analysis:** Elizabeth K. Serafin, Judy J. Yoo.

**Funding acquisition:** Mark L. Baccei.

**Investigation:** Elizabeth K. Serafin, Judy J. Yoo, Jie Li.

**Methodology:** Elizabeth K. Serafin, Jie Li, Xinzhong Dong, Mark L. Baccei.

**Project administration:** Elizabeth K. Serafin, Mark L. Baccei.

**Resources:** Xinzhong Dong, Mark L. Baccei.

**Supervision:** Elizabeth K. Serafin, Mark L. Baccei.

**Writing – original draft:** Elizabeth K. Serafin.

**Writing – review & editing:** Elizabeth K. Serafin, Judy J. Yoo, Xinzhong Dong, Mark L. Baccei.

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
