## [Decision Letter · Decision Letter 0]

21 Nov 2023

PONE-D-23-33257Development and characterization of a *Gucy2d-cre* mouse to selectively manipulate a subset of inhibitory spinal dorsal horn interneuronsPLOS ONE

Dear Dr. Serafin,

Thank you for submitting your manuscript to PLOS ONE. After careful consideration, we feel that it has merit but does not fully meet PLOS ONE’s publication criteria as it currently stands. Therefore, we invite you to submit a revised version of the manuscript that addresses the points raised during the review process.

 The following points of the reviewers require special attention and need to be addressed:-  Was an immunostaining for a molecular marker of Gucy2d+ olfactory neurons performed? If not, explain why.- Concerning Fig. 3 and 4, please clarify the issues upon reporter and Gucy2d expression.- Provide check for leaky expression of tdTomato or SunGFP1 and report what control animals were used in the experiments with AAV injection.

We look forward to receiving your revised manuscript.

Kind regards,

Karl-Wilhelm Koch, Ph.D.

Academic Editor

PLOS ONE

 [All work was supported by NS100469,

NIH National Institute of Neurological Disorders and Stroke (https://www.ninds.nih.gov/), awarded to MLB.].  

4. We note that Figure(s) 1, 2, 3, 4 and 5 in your submission contain copyrighted images. All PLOS content is published under the Creative Commons Attribution License (CC BY 4.0), which means that the manuscript, images, and Supporting Information files will be freely available online, and any third party is permitted to access, download, copy, distribute, and use these materials in any way, even commercially, with proper attribution. For more information, see our copyright guidelines: http://journals.plos.org/plosone/s/licenses-and-copyright.

a. You may seek permission from the original copyright holder ofFigure(s) 1, 2, 3, 4 and 5 to publish the content specifically under the CC BY 4.0 license. 

Reviewers' comments:

Reviewer's Responses to Questions

**Comments to the Author**

1. Is the manuscript technically sound, and do the data support the conclusions?

Reviewer #1: Partly

Reviewer #2: Yes

2. Has the statistical analysis been performed appropriately and rigorously? 

Reviewer #1: N/A

Reviewer #2: N/A

3. Have the authors made all data underlying the findings in their manuscript fully available?

Reviewer #1: Yes

Reviewer #2: Yes

4. Is the manuscript presented in an intelligible fashion and written in standard English?

Reviewer #1: Yes

Reviewer #2: Yes

5. Review Comments to the Author

Reviewer #1: In this study, Serafin and colleagues generated a Gucy2d-Cre mouse line to facilitate the manipulation of inhibitory dynorphin interneurons in the mouse spinal dorsal horn. They used CRISPR/Cas9 to knock in the Cre recombinase into the endogenous Gucy2d locus of single-cell C57BL/6J embryos. Founder (F0) mice were bred with two Cre-dependent reporter lines, and double-heterozygous offspring showed reporter labeling of a larger-than-expected population of inhibitory neurons, which included spinal and brain regions where Gucy2d expression has not been reported. The confounding effect of transient Gucy2d expression during development was discarded after analyzing adult Gucy2d-Cre mice injected with AAV-CAG-FLEX-tdTomato viral vectors. The authors suggest that Gucy2d expression at levels below detection, yet sufficient to induce Cre-dependent reporter, accounts for the unexpected cell labeling. They also discuss the limitations and suitability of using Gucy2d-Cre mice to target a wide population of inhibitory interneurons, and Gucy2d+ olfactory neurons.

The authors provide a valid justification for generating a new Cre-driver strain, and the choice of the Gucy2d locus seems well-grounded. Although the Gucy2d-Cre strain might not be ideal for the purpose it was initially designed for, it might still be useful in other contexts. It is not uncommon that Cre-driver strains exhibit inconsistent and off-target Cre activity, and I believe that expanding the characterization of Cre expression in Gucy2d-Cre mice could increase the reliability of this strain and help explain the unexpected results. Moreover, I would like to suggest a few modifications to the figures that would make information more readily available to the readers. My questions and comments are listed below.

1) Fig. 1A – A scheme showing the genomic target site (Gucy2d gene, with introns and exons), Cre-donor vector (indicating the sites where homology recombination is expected), and the resulting knock-in allele, would convey more details of the integration strategy.

2) Fig. 1B – Did the authors test whether unexpected cell labeling also occurred in the olfactory epithelium? Immunostaining for a molecular marker of Gucy2d+ olfactory neurons (such as phosphodiesterase 2A (PDE2A), for example) could help address this question.

3) Fig. 3 – Using arrows (or another symbol) to point out signal colocalization would be really helpful. In Fig. 3C, it is especially difficult to visualize cyan-only (Gucy2d) and cyan + green (Gucy2d + Gucy2d-Sun1GFP) cells. Alternatively, another panel where the different color channels are shown separately could be added. Moreover, it is mentioned in the main text that Gucy2d mRNA was localized to laminae I-II, and a substantial number of Gucy2d-Sun1GFP nuclei was present in lamina III-V; could the authors delimitate these laminae in the figure?

4) Fig. 4 – It is mentioned that reporter expression is clearly present in laminae III-V of the spinal dorsal horn of Gucy2d-Cre x AAV8-FLEX-tdTom animals, however, only laminae I-III are indicated. Moreover, does this reporter expression differ from the observed in Gucy2d x Ai9 animals?

5) Fig. 5 –The similarities between the top (A-D) and bottom (E-H) panels are not clearly visible, probably due to magnification and background differences. Could the authors point out in the images what they are trying to convey?

6) Were the same founders bred with the two reporter lines? In my understanding, Figs 2a and 3a are representative images of, respectively, tdTomato and SunGFP1 labeling in the SDH, but it seems like there are substantially fewer positive cells in 3a. What were the overall differences (if any) observed in labeling with each reporter?

7) What controls were used to check for leaky expression of tdTomato or SunGFP1? And what control animals were used in the experiments with AAV injection?

8) Was Cre expression evaluated (by using ISH or IF, for example)?

9) It is intriguing that approximately 60% of Gucy2d-Sun1GFP cells in the SDH were not labeled with either Gucy2d or Pdyn ISH probes. What is the feasibility of sorting Sun1GFP+ cells for an RNA-sequencing analysis? Can the authors discuss some experimental strategies that would help determine the identity of these cells?

Reviewer #2: The manuscript by Serafin et al, describes the generation and characterization of a new Gucy2d-cre transgenic mouse line. The manuscript explains that these mouse lines can be used to target specific populations of spinal cord neurons as well as neurons in supraspinal regions of the CNS. The experiments were well performed, and the claims of the study are all reasonable. I have a few minor comments on the texts which will only require minor revisions.

1. It is not clear whether all founders (the three that bred) were analyzed. Although not generally acknowledged in many publications, different CRIPR knockin founders can behave differently from each other with different expression patterns (similar to BAC transgenic mice).

2. The paper cites the study Boyle et al to describe the mixed inhibitory and excitatory populations of dynorphin neurons. The MS should also cite Huang et al, 2018 Nature Neurosci as this study describes this population further as well as showing the functional consequences of only activating dynorphin neurons (albeit the mixed excitatory and inhibitory classes).

3. Related to point 2. As detailed in Huang et al, 2018, excitatory neurons are found in the lumbar and cervical enlargements (with a medial location) and are not present in thoracic segments.

4. Why was TgN expression not expected? Pydn-neurons are present in the spinal and parts of the interpolar TgN.

5. Lines 311-116. An additional comment should be added to the description of the Lbx1 and Pdyn intersectional approach used by the Goulding/Ma labs. Namely, this approach likely captures lineages of neurons which extend to ablation of neurons beyond the adult expression of dynorphin.

6. Line 336, sentence needs addition of a “the”.

6. PLOS authors have the option to publish the peer review history of their article (what does this mean?). If published, this will include your full peer review and any attached files.

Reviewer #1: No

Reviewer #2: **Yes: **Mark Hoon

---

## [Author Response · Author response to Decision Letter 0]

20 Feb 2024

Dear Dr. Chenette,

We would like to thank the reviewers for their constructive feedback on our manuscript (MS) entitled “Development and characterization of a Gucy2d-cre mouse to selectively manipulate a subset of inhibitory spinal dorsal horn interneurons.” Their comments have resulted in both new experimental data and textual changes to the manuscript (as shown in the Track Changes copy) as well as updated figures (Figures 1, 3, and 5) and the addition of two Supplemental Figures (S1 Fig and S2 Fig). We respond to all comments on a point-by-point basis below:

Reviewer 1:

1) Fig. 1A – “A scheme showing the genomic target site […], Cre-donor vector […], and the resulting knock-in allele would convey more details of the integration strategy.”

We have updated Fig. 1A to a more detailed schematic. However, the Gucy2d gene is very large and contains 19 exons, so we only showed the portion of the gene where integration occurs.

2) Fig. 1B – Did the authors test whether unexpected cell labeling also occurred in the olfactory epithelium? Immunostaining for a molecular marker of Gucy2d+ olfactory neurons (such as phosphodiesterase 2A (PDE2A), for example) could help address this question.

We did not perform immunostaining to conclusively identify Gucy2d+ olfactory neurons in the MOE, as this was deemed outside the scope of our study which focused primarily on characterization of lumbar spinal cord expression. As such, we have avoided calling these labeled cells in the MOE “GC-D+ neurons” and only noted that their location appears consistent with prior literature describing this population.

3) Fig. 3 – Using arrows (or another symbol) to point out signal colocalization would be really helpful. In Fig. 3C, it is especially difficult to visualize cyan-only (Gucy2d) and cyan + green (Gucy2d + Gucy2d-Sun1GFP) cells. 

We agree and have revised Fig. 3 to better show signal colocalization, or lack thereof, especially in Fig. 3C. To improve differentiation between two-color colocalization and three-color colocalization, we also changed the colors of the Gucy2d and Pdyn fluorescent channels. Additionally, we added arrowheads and other symbols to point out examples of colocalized cells as described in the revised figure legend. 

4) Fig. 4 – It is mentioned that reporter expression is clearly present in laminae III-V of the spinal dorsal horn of Gucy2d-Cre x AAV8-FLEX-tdTom animals, however, only laminae I-III are indicated. Moreover, does this reporter expression differ from the observed in Gucy2d x Ai9 animals?

We have changed our wording in the MS to omit references to specific laminae beyond I-III; we now refer to “deeper dorsal horn laminae” or similar since expression outside laminae I-III was not characterized in detail. Cre-dependent reporter expression in laminae I-III appears similar across both the Gucy2d-cre x Ai9 breeding approach and the Gucy2d-cre AAV-FLEX-tdTomato approach insofar as labeling in lamina III was not abolished when expression is driven acutely by a cre-dependent reporter vector, indicating that this labeling is not the result of transient developmental expression of cre in these cells. We have adjusted the wording in the Results (line 270) and figure legend (line 280) to make this point clearer.

5) Fig. 5 –The similarities between the top (A-D) and bottom (E-H) panels are not clearly visible, probably due to magnification and background differences. 

We have added labels to each panel to more clearly indicate brain regions which exhibit cre-dependent reporter expression with both the Gucy2d-cre x Ai9 breeding approach and the injection of systemic AAV-PHP.eB-FLEX-tdTomato into Gucy2d-cre mice. We have also added labels for landmarks such as ventricles to facilitate orientation. Panel 5C has also been exchanged for a different image so that we could show the same cortical area in both Figs. 5C and 5G.

6) Were the same founders bred with the two reporter lines? In my understanding, Figs 2a and 3a are representative images of, respectively, tdTomato and SunGFP1 labeling in the SDH, but it seems like there are substantially fewer positive cells in 3a. What were the overall differences (if any) observed in labeling 

Both reviewers requested more details regarding which founder(s) and their progeny were analyzed and presented in the study, and correctly pointed out that progeny of different founders could exhibit differences in reporter expression. It is important to note that all of the data presented in this manuscript are from one single founder and its progeny. We did analyze the progeny of 3 additional potential founders, 2 of which exhibited no noticeable differences in reporter expression in the spinal cord, DRG, or brain compared to the founder we ultimately moved forward with. The remaining potential founder was excluded due to inconsistent expression as described in the text. We have clarified this issue in both the Methods (lines 123-126) and Results (lines 188-189) sections, which now specify that all images and data are from a single founder and its progeny.

We would also note that the reduced number of positive cells in the Sun1GFP-labeled spinal cord images compared to the tdTomato images is due to the reduced thickness of tissue sections prepared for RNAscope or NeuN immunohistochemistry (Sun1GFP; 14 μm) compared to those prepared for gross anatomical characterization (tdTomato; 25 μm). As a result, fewer cells are captured per section in the Sun1GFP tissue.

7) What controls were used to check for leaky expression of tdTomato or SunGFP1? And what control animals were used in the experiments with AAV injection?

Checks for leaky expression were performed using cre-negative Ai9 mice or cre-negative Sun1GFP mice, and images of these controls are now included in Supplemental Fig. 1 (S1 Fig). Additionally, AAV-FLEX tdTomato vectors were injected systemically (AAV-PHP.eB, to assess brain expression) or intraspinally (AAV8) to cre-negative mice to assess potential cre-independent (i.e., leaky) expression using a viral-driven reporter. These control images are shown as Supplemental Fig. 2 (S2 Fig), and the number of mice used for each of these experiments is provided in the Methods. In summary, leaky expression was not detected in any of these control experiments.

8) Was Cre expression evaluated (by using ISH or IF, for example)?

Since Ai9 and Sun1GFP mice showed no leaky expression, this was not evaluated directly.

9) […]What is the feasibility of sorting Sun1GFP+ cells for an RNA-sequencing analysis? Can the authors discuss some experimental strategies that would help determine the identity of these cells?

Yes, such an experiment would certainly be feasible, as we and others have sorted Sun1GFP+ nuclei for both population-level and single-nucleus RNAseq analysis (Chamessian et al., Sci Rep 2019; Serafin et al., PAIN 2019; Serafin et al., PAIN 2021). Although we decided against performing this experiment for time and cost reasons, we have suggested this as a future direction in the Discussion (lines 366-372).

Reviewer 2:

1) It is not clear whether all founders (the three that bred) were analyzed. Although not generally acknowledged in many publications, different CRISPR knockin founders can behave differently from each other with different expression patterns (similar to BAC transgenic mice).

Please see the above response to Point #6 from Reviewer 1.

2. The paper cites the study Boyle et al to describe the mixed inhibitory and excitatory populations of dynorphin neurons. The MS should also cite Huang et al, 2018 Nature Neurosci as this study describes this population further as well as showing the functional consequences of only activating dynorphin neurons (albeit the mixed excitatory and inhibitory classes).

and

3. Related to point 2. As detailed in Huang et al, 2018, excitatory neurons are found in the lumbar and cervical enlargements (with a medial location) and are not present in thoracic segments.

We agree it was an oversight not to cite Huang et al. 2018 and have added relevant information from this study to our Introduction (lines 64-66). 

4. Why was TgN expression not expected? Pydn-neurons are present in the spinal and parts of the interpolar TgN.

We agree and have rephrased our description of brain expression in the Results (lines 284-286) and Discussion to better differentiate between the expected expression observed in the TgN (Sp5) and the unexpected expression in other brain areas such as the thalamus and cortex. 

5. Lines 311-116. An additional comment should be added to the description of the Lbx1 and Pdyn intersectional approach used by the Goulding/Ma labs. Namely, this approach likely captures lineages of neurons which extend to ablation of neurons beyond the adult expression of dynorphin.

We agree and have expanded on the Goulding/Ma approach in the Discussion (lines 334-336) as suggested. A typo in line 336 (now 360) of the Discussion has also been corrected.

Having addressed the reviewer comments as described above, we hope that our manuscript now meets the criteria for publication in PLOS One. We have also verified that the manuscript adheres to PLOS One editorial requirements including file naming, financial disclosure and ethics statements (included in Methods). To this end, we provide the amended Role of Funder statement: “The funders had no role in study design, data collection and analysis, decision to publish, or preparation of the manuscript.” We also affirm that the figures in this submission are author-created and have not been published anywhere else. Thus, they adhere to PLOS One’s copyright policy.

Please feel free to contact me at kritzeee@ucmail.uc.edu for any further discussion about this submission. Thank you for your time, and we look forward to hearing from you soon.

Best wishes,

Elizabeth K. Serafin, M.S.

Principal Research Assistant

Pain Research Center, University of Cincinnati

---

## [Editor Report · Decision Letter 1]

26 Feb 2024

Development and characterization of a *Gucy2d-cre* mouse to selectively manipulate a subset of inhibitory spinal dorsal horn interneurons

PONE-D-23-33257R1

Dear Dr. Serafin,

We’re pleased to inform you that your manuscript has been judged scientifically suitable for publication and will be formally accepted for publication once it meets all outstanding technical requirements.

Kind regards,

Karl-Wilhelm Koch, Ph.D.

Academic Editor

PLOS ONE
---

## [Editor Report · Acceptance letter]

5 Mar 2024

PONE-D-23-33257R1 

PLOS ONE

Dear Dr. Serafin, 

I'm pleased to inform you that your manuscript has been deemed suitable for publication in PLOS ONE. Congratulations! Your manuscript is now being handed over to our production team.

Kind regards, 

on behalf of

Dr. Karl-Wilhelm Koch 

Academic Editor

PLOS ONE